# Cartilage Regeneration Potential in Early Osteoarthritis of the Knee: A Prospective, Randomized, Open, and Blinded Endpoint Study Comparing Adipose-Derived Mesenchymal Stem Cell (ADSC) Therapy Versus Hyaluronic Acid

**DOI:** 10.3390/ijms26178476

**Published:** 2025-08-31

**Authors:** Ponthep Tangkanjanavelukul, Saradej Khuangsirikul, Danai Heebthamai, Montarop Yamabhai, Thitima Sumphanapai, Nattapat Khumtong, Thanainit Chotanaphuti

**Affiliations:** 1Department of Orthopedics, School of Medicine, Institute of Medicine, Suranaree University of Technology, 111 University Avenue, Nakhon Ratchasima 30000, Thailand; 2Department of Orthopaedic, Phramongkutklao College of Medicine, 317 Ratchawithi Road, Thung Phaya Thai, Ratchathewi, Bangkok 10400, Thailand; ksaradej@yahoo.com (S.K.); danaiheeb@hotmail.com (D.H.); 3Molecular Biotechnology Laboratory, School of Biotechnology, Institute of Agricultural Technology, Suranaree University of Technology, 111 University Avenue, Nakhon Ratchasima 30000, Thailand; montarop@g.sut.ac.th; 4Division of Clinical Hematology and Microscopy, Department of Medical Technology, School of Allied Health Sciences, University of Phayao, Phayao 56000, Thailand; thitima.su@up.ac.th; 5Department of Family Medicine, School of Medicine, Institute of Medicine, Suranaree University of Technology, 111 University Avenue, Nakhon Ratchasima 30000, Thailand; nattapat.tanwa@gmail.com

**Keywords:** knee osteoarthritis, adipose-derived mesenchymal stem cells, stem cell therapy, regenerative medicine

## Abstract

Early-stage knee osteoarthritis (knee OA) lacks effective regenerative therapies. This study aimed to compare the cartilage regenerative effects, clinical efficacy, and safety of intra-articular injections of autologous adipose-derived mesenchymal stem cells (ADSCs) versus hyaluronic acid (HA). Forty-eight patients with early knee OA were enrolled in a prospective open-blinded multi-center study at Suranaree University of Technology Hospital and Phramongkutklao Hospital. Participants were randomized into either the ADSC or HA group. Primary outcomes included MRI-based cartilage lesion volume, synovial thickness via ultrasound, and WOMAC scores over 6 months. MRI results revealed significant and progressive cartilage regeneration in the ADSC group. In particular, medial femoral cartilage lesion volume decreased by 50.06 mm^3^, whereas the HA group showed an increase of 36.44 mm^3^. Synovial thickness also declined significantly in the ADSC group at 3 and 6 months. Both groups demonstrated reduced symptoms, but the ADSC group achieved superior and sustained improvements in WOMAC pain, stiffness, and function scores throughout the 6-month follow-up. The clinical benefits were consistent and more pronounced compared with HA. No serious adverse events occurred. In conclusion, intra-articular ADSC injections show superior cartilage restoration on MRI and better clinical outcomes than HA injection, making them a promising treatment for early-stage knee OA.

## 1. Introduction

Osteoarthritis (OA) is a degenerative and inflammatory joint disease that causes significant pain and disability in the elderly. Knee OA is more prevalence when compared with other types of OA [1] and presents at an earlier age in women than men [2]. The global incidence of knee OA was 203 per 10,000 person per year [3]. Symptomatic knee OA is likely to increase due to aging society and the obesity epidemic. A number of treatment options for knee OA are available. The standard treatments include traditional pharmacological treatments, non-pharmacological treatments, and surgical procedures. However, these procedures can only improve disease-related symptoms but not effectively repair the damaged cartilage.

In the early stages, when cartilage integrity is relatively preserved, non-operative treatments remain central to management. Intra-articular hyaluronic acid (HA) has been widely used as a visco-supplementation therapy to restore synovial fluid viscoelasticity, improve lubrication, and reduce joint stress [4].

The role of HA, however, remains debated due to variable clinical outcomes. The OARSI 2019 guidelines provide a balanced position, categorizing HA as a conditional option (Level 1B/2) appropriate in selected patients, particularly when comorbidities limit systemic therapies [5]. Evidence from randomized controlled trials [4] and meta-analyses [6,7] indicate that optimized HA formulations, especially high molecular weight or cross-linked preparations, are more effective in patients with early-stage OA, where cartilage preservation and lower inflammation may enhance responsiveness. Thus, while not universally endorsed, HA retains a role in carefully selected patients, forming part of a tailored management strategy alongside emerging biologic therapies such as mesenchymal stem cells (MSCs).

Due to recent advancements in cellular regenerative therapy, MSCs have emerged as an alternative treatment for multiple diseases, including knee OA. In addition to effectively relieving pain and improving motor function of patients with knee OA [8], intra-articular injection of mesenchymal stem cells has been reported to significantly improve function and quality of cartilage by T2-weighted magnetic resonance imaging (T2 MRI) [9]. Long-term feasibility, safety, and clinical efficacy of knee OA treatment by using autologous bone marrow MSCs have been reported [10]. However, the isolation of bone marrow-derived MSCs has some impediments because of invasive and complicated harvesting procedures. To overcome this difficulty, adipose-derived mesenchymal stem cells (ADSCs) are an attractive source of MSCs. The advantages of ADSCs over bone marrow MSCs include lower patient morbidity, easier accessibility, and higher yield. In addition, ADSCs have been shown to have a higher angiogenic potential, promoting the formation of new blood vessels. These crucial roles of ADSCs contribute to tissue regeneration and repair [11]. Several randomized controlled trials (RCTs) have investigated the role of ADSCs in knee OA. Despite the use of stem cell therapy, prior studies often enrolled heterogeneous patient populations comprising both early and advanced stage knee osteoarthritis, which may have limited the ability to detect consistent therapeutic effects. Although clinical outcomes improved, significant structural changes were not consistently observed. This lack of significant structural regeneration may be due to the inclusion of patients with severe osteoarthritis, where the extent of degeneration is too advanced for effective regeneration [12,13].

While many studies on the clinical efficacy of ADSCs in patients with severe knee OA [14] have been reported, RCTs focusing on the clinical efficacy of using ADSCs in patients with early-stage knee OA have not been widely investigated. Early-stage knee OA is also an interesting phase for treatment with cellular regenerative medicine. Studies on the use of ADSCs in patients with early-stage knee osteoarthritis are currently limited. Therefore, it is important to focus research on this patient group to achieve cartilage regeneration, including investigating injection methods, protocols, and cell dosages.

To evaluate the primary outcome, magnetic resonance imaging (MRI) and ultrasonography (US) have been widely used to assess knee structural and morphological changes [12,15,16]. In addition, for secondary outcomes, including pain and functional changes, the Western Ontario and McMaster Universities Osteoarthritis Index (WOMAC) is widely used. The use of urinary C-terminal cross-linked telopeptide of type II collagen (urinary CTX-II), a potential OA biomarker, for monitoring cartilage degradation in patients with knee OA has also been reported [17]. Interestingly, a comprehensive assessment combining MRI, US, and urinary CTX-II in a single study has never been established. Here, we conducted a randomized, prospective, open, and blinded endpoint study to compare clinical efficacy, cartilage imaging, safety profile, and biological marker for cartilage degradation between intra-articular injection of autologous ADSCs versus hyaluronic acid (HA) (active control group), among patients with early-stage knee OA.

This study aims to assess the efficacy of ADSC therapy in early-stage knee OA by comparing it with HA therapy. Does ADSC therapy lead to (1) a significant improvement in cartilage regeneration as assessed by MRI, (2) a greater reduction in synovial thickness, and (3) better pain as measured by WOMAC scores? (4) Are there any significant adverse events associated with ADSC therapy?

## 2. Results

A total of 48 participants met the inclusion criteria and were enrolled in this study. They were randomly allocated into 2 groups: 24 participants (51.06%) were assigned to the HA (control) group, and 23 participants (48.94%) to the ADSC (intervention) group. In terms of sex distribution, each group comprised 4 male participants, while the control group had 20 female participants and the intervention group had 19. The mean age in the HA group was 59.00 ± 6.69 years, compared with 56.91 ± 6.15 years in the ADSC group. Average body weight was comparable between groups, measuring at 61.23 ± 8.90 kg in the control group and 61.73 ± 8.62 kg in the intervention group. These demographic variables did not differ significantly between groups (Table 1).

Baseline clinical and imaging parameters were well balanced between the two groups. The average baseline WOMAC score was slightly higher in the HA group (96.46 ± 38.98) than in the ADSC group (88.26 ± 35.21), suggesting marginally more severe symptoms in the control group prior to treatment. Pre-intervention ultrasound measurements of synovial membrane thickness showed a mean of 3.25 ± 0.6 mm in the HA group and 3.70 ± 0.7 mm in the ADSC group. Although the intervention group exhibited slightly thicker synovial membranes at baseline, the difference was not statistically significant. Similarly, baseline urinary CTX-II levels, an established biomarker of cartilage degradation, were 310.45 ng/mmol in the HA group and 263 ng/mmol in the ADSC group, without a statistically significant difference (Table 1).

Taken together, these findings confirm that both groups were comparable at baseline in terms of demographic, clinical, and imaging characteristics. However, the slightly thinner synovial membrane in the HA group may suggest a lower baseline level of synovial inflammation compared with the ADSC group, which could influence early post-treatment responses.

### 2.1. Cartilage Regeneration and Structural Changes

MRI evaluations over the 6-month period revealed progressive cartilage regeneration in the ADSC group, particularly in the medial femoral region, an effect that was not observed in the HA group. Grade 1 cartilage lesion (6 × 9 mm) at the mid-lateral femoral condyle disappeared, with a decrease in the focal grade mid-femoral trochlea observed on the axial view (Figure 1A) and sagittal view (Figure 2A). Notable cases included the complete disappearance of a Grade 1 cartilage lesion at the mid-lateral femoral condyle and a reduction in lesion size from Grade 3 (16 × 12 mm) to Grade 2 (15 × 11 mm) at the medial femoral condyle (Figure 1B and Figure 2B). A Grade 1 lesion at the medial patella facet (4 × 5 mm) was also reduced in size, while areas of Grade 3 cartilage lesions at the medial femoral condyle (14 × 9 mm) and medial tibial plateau (10 × 6 mm) showed slight reductions) (Figure 2C). In contrast, the MRI findings in the group treated with intra-articular hyaluronic acid injections demonstrated that the cartilage lesion areas either remained unchanged or showed signs of progression (Appendix A).

Moreover, our findings demonstrated that ADSC therapy resulted in substantial cartilage regeneration, an effect not observed in the HA group. Specifically, the lesion area in the medial femoral cartilage of the ADSC group decreased by an average of 36.44 mm^3^, whereas in the HA group it increased by 50.06 mm^3^ (Figure 3A). Similarly, the lesion in the medial patella cartilage decreased by 37.91 mm^3^ in the ADSC group but increased by 10.7 mm^3^ in the HA group (*p* < 0.05) (Figure 3B). These results strongly suggest that ADSC therapy effectively contributes to cartilage preservation and regeneration, providing structural benefits beyond symptom relief. The lesion area in the medial femoral cartilage of the ADSC group decreased by an average of 15.06 mm^2^, whereas in the HA group it increased by an average of 9.62 mm^2^ (Figure 3C). However, this difference was not statistically significant (*p* = 0.166). The lesion area in the medial patella cartilage of the ADSC group decreased by an average of 17.5 mm^2^, whereas in the HA group it increased by an average of 5.07 mm^2^, with this difference being statistically significant (*p* < 0.05) (Figure 3D). Over the 6-month period, patients in the control group demonstrated mild progression in cartilage damage, with increases in lesion area of 6.42 mm^2^ at the medial femoral condyle and 3.17 mm^2^ at the medial patella. In contrast, patients treated with ADSCs showed a further reduction in cartilage lesion area 18.20 mm^2^ at the femoral site (*p* = 0.103) and a statistically significant 12.30 mm^2^ at the patella site (*p* < 0.05), suggesting superior surface preservation. Similarly, cartilage lesion volume in the control group increased by 33.38 mm^3^ (femoral) and 6.29 mm^3^ (patella), while the ADSC group exhibited additional reductions of 61.90 mm^3^ (*p* < 0.05) and 26.27 mm^3^ (*p* < 0.05), respectively (Appendix A). These findings indicate that intra-articular ADSC therapy may not only slow cartilage degeneration but also significantly reduce lesion size, especially at the patella site, supporting its potential therapeutic value in early-stage knee osteoarthritis.

### 2.2. Reduction in Synovial Thickness

Synovial membrane thickness was used as an indicator of joint inflammation, and changes were monitored over 6 months. In the ADSC group, synovial thickness was 3.70 mm before injection, decreasing to 3.04 mm at 1 month and further to 2.75 mm at 3 months. By the 6-month follow-up, there was a slight increase to 2.86 mm, but it remained lower than baseline values (Table 2). In contrast, the HA group exhibited an initial decrease in synovial thickness from 3.25 mm to 3.09 mm at 1 month, followed by an increase to 3.19 mm at 3 months, and a further significant increase to 3.96 mm at 6 months (*p* < 0.05). These findings indicate that, while the HA group initially experienced a temporary reduction in inflammation, it tended to return over time. Conversely, in the ADSC group, synovial membrane thickness decreased rapidly during the first 3 months and remained stable thereafter, suggesting a more sustained anti-inflammatory effect of ADSC therapy (Figure 4A–C).

### 2.3. Clinical Outcomes and WOMAC

In terms of clinical outcomes, both ADSC and HA therapy resulted in a decrease in WOMAC scores, indicating improvements in pain, stiffness, and function. In the ADSC group, the overall WOMAC score significantly improved from 88.26 to 26.30, while in the HA group it improved from 96.46 to 49.09 (Table 3). A subgroup analysis revealed that the WOMAC pain score in the ADSC group showed a progressive decline from a baseline of 19.91 to 18.35 at 1 month, 11.83 at 3 months, and 5.83 at 6 months. In contrast, the HA group exhibited an initial improvement from 21.92 to 13.63 at 1 month, followed by a slight increase to 15.38 at 3 months, before declining again to 11.18 at 6 months (Table 3). A similar trend was observed for WOMAC stiffness scores, where the ADSC group showed continuous improvement from a baseline of 7.83 to 1.35 at 6 months, while the HA group exhibited a modest reduction from 8.17 to 4.68 at 6 months (Table 3). Likewise, WOMAC function scores improved more consistently in the ADSC group, decreasing from 60.52 to 19.13 at 6 months, compared with the HA group, which improved from 69.14 to 33.23 over the same period (Table 3). These results indicate that while both ADSC and HA therapy provided symptomatic relief, ADSC therapy resulted in more sustained and consistent improvements across all WOMAC components (Figure 5A–D).

Further statistical analysis demonstrated that the mean difference in WOMAC scores between the ADSC and HA groups was 22.79 points, which was statistically significant. Subgroup differences showed a significant improvement in function scores (14.1 points), while pain and stiffness scores improved by 5.35 and 3.33 points, respectively. Notably, ADSC therapy demonstrated a stronger trend toward improvement in the overall WOMAC score, with particularly significant reductions in stiffness and function scores at the 3- and 6-month follow-ups.

### 2.4. Biomarker Analysis

Urine CTX-II levels were measured as an exploratory biomarker. No significant difference was observed between groups at 6 months (*p* = 0.881). The ADSC group showed a mean change of −82 ng/mmol (95%CI −219.0 to +119.0 ng/mmol), while the HA group showed a mean change of −69.63 ng/mmol (95%CI −214.44 to +132.36 ng/mmol) (Table 4). Given the wide confidence intervals and small sample size, further studies are needed to clarify potential biomarker changes.

### 2.5. Safety and Adverse Events

Regarding safety outcomes, no serious adverse events were reported in either group. In the ADSC group, one participant (4.34%) experienced mild knee effusion and pain five days after the injection, but these symptoms resolved quickly with conservative treatment, including cold compression and rest. No long-term complications were observed, indicating that ADSC therapy was well-tolerated and had a favorable safety profile.

## 3. Discussion

Knee osteoarthritis (knee OA) is a progressive degenerative disease, and stem cell therapy has emerged as a promising treatment option. However, previous studies on stem cell injections for knee OA have often failed to distinguish between early-stage and late-stage knee OA, leading to variations in study populations and making it challenging to draw definitive conclusions on efficacy. Additionally, differences in stem cell characteristics, types, and preparation methods significantly influence treatment outcomes. While multiple studies have suggested that stem cell therapy can improve symptoms in patients with knee OA [18,19,20,21], uncertainty remains as to whether these improvements translate into statistically significant structural changes, particularly in cartilage regeneration.

Autologous uncultured stem cells offer the advantage of utilizing a patient’s own cells, reducing the risk of immune rejection. However, controlling the quality and quantity of autologous cells presents challenges, making it difficult to achieve consistent results. In contrast, allogeneic stem cells introduce additional variables that may further impact treatment outcomes. In this study, we specifically utilized autologous cultured adipose-derived stem cells (ADSCs) at a standardized dose of 50 million cells, ensuring precise delivery through ultrasound-guided injections. This approach enhances treatment consistency, improves reproducibility of results, and optimizes the potential for cartilage regeneration and symptom relief in patients with early-stage knee OA.

Our study aimed to evaluate the efficacy of autologous ADSCs in promoting cartilage regeneration in early-stage knee OA compared with hyaluronic acid (HA) therapy. We observed that the lesion area in the medial femoral cartilage of the ADSC group decreased by an average of 15.06 mm^2^, whereas in the HA group it increased by 9.62 mm^2^; however, this difference was not statistically significant (*p* = 0.166) (Figure 3C). Conversely, the lesion area in the medial patella cartilage of the ADSC group decreased by an average of 17.5 mm^2^, while in the HA group it increased by 5.07 mm^2^, with this difference reaching statistical significance (*p* = 0.024) (Figure 3D).

These findings align with the existing literature, suggesting that ADSC therapy may offer structural benefits in cartilage preservation and regeneration. For instance, a study utilizing multi-compositional MRI sequences demonstrated that allogeneic human adipose-derived mesenchymal progenitor cells (haMPCs) promoted cartilage repair, as evidenced by compositional alterations indicative of regeneration [22]. Additionally, a clinical phase I/II trial investigating stromal vascular fraction (SVF) therapy for cartilage regeneration reported positive outcomes, further supporting the potential of adipose-derived stem cells in treating knee OA [23,24].

Moreover, a critical review of clinical trials involving mesenchymal stem cell (MSC) therapies for OA highlighted their efficacy in cartilage regeneration, although it emphasized the need for robust clinical trials to generate reliable evidence supporting these treatments. Collectively, these studies, along with our findings, suggest that ADSC therapy holds promise in enhancing structural cartilage outcomes in patients with early-stage knee OA.

Another key objective was to evaluate whether ADSC therapy results in a greater reduction in synovial thickness than HA therapy. In our study, we observed that the mean synovial thickness in the ADSC group remained consistently lower over a 12-week period compared with the HA group. Notably, at the 72-week mark, the ADSC group exhibited a mean synovial thickness that was 0.84 mm less than pre-injection levels, whereas the HA group showed an increase of 0.71 mm above pre-intervention levels, resulting in a statistically significant difference (Table 2). This finding aligns with previous research indicating that intra-articular corticosteroid injections maintain reduced synovial thickness over a 12-week follow-up, while hyaluronic acid injections initially reduce synovial thickness in the first 4 weeks, followed by a rebound to pre-injection levels thereafter [25].

In our study, ADSC therapy resulted in significantly greater and more sustained improvements in WOMAC pain, stiffness, and function scores compared with HA. While both groups showed clinical improvement, the ADSC group demonstrated a consistently stronger trend across all time points, particularly in the function and stiffness subscales. These findings align with previous studies, such as that by Jo et al. [19], who reported an decrease in WOMAC scores in the 10 million cell group at 3 months. Another study also observed an improvement in functional scores, specifically knee injury and osteoarthritis outcome score (KOOS) at 12 months [26,27]. Additionally, meta-analysis studies support our conclusion that ADSC therapy provides greater clinical improvement than HA, showing a statistically significant mean WOMAC difference of 22.79, further confirming the lasting benefits of ADSC treatment [19,27]. Overall, ADSC therapy appears to offer superior clinical benefits over HA for early-stage knee osteoarthritis. However, some studies reported improvements in WOMAC scores over 6 months, although the results did not reach statistical significance [13].

Safety remains a key consideration in evaluating new therapies. In this study, post-implantation pain was reported in only 1 out of 23 cases (4.34%), presenting as mild knee pain and swelling after injection. This incidence was significantly lower compared with moderate- to high-dose allogeneic MSC therapy, where post-implantation pain has been observed in approximately 53% to 60% of patients in both the experimental and control groups [21]. These findings suggest that properly prepared autologous ADSCs result in a lower incidence of adverse effects compared with allogeneic MSCs administered at the same dosage [18,28].

This study has some limitations that should be acknowledged. While the sample size was sufficient to detect meaningful differences between groups, larger studies are necessary to confirm the long-term effects of ADSC therapy. Additionally, variations in cartilage degeneration patterns among patients may have influenced individual responses to treatment. Although ultrasound-guided injections ensured precision in cell delivery, anatomical differences and variations in baseline synovial thickness may have contributed to outcome variability. Future studies should aim for a larger cohort and extended follow-up periods to validate these findings.

## 4. Materials and Methods

### 4.1. Patients and Procedures

We enrolled 48 participants who had been diagnosed with symptomatic knee osteoarthritis Kellgren & Lawrence stage II in a multi-center study conducted at Suranaree University of Technology Hospital and Phramongkutklao Hospital. These participants were between the ages of 40 and 70 and were experiencing moderate pain associated with mild to moderate knee osteoarthritis (knee OA). The sample size was calculated according to the formula [29] (Appendix A).

Exclusion criteria were applied to individuals who had a history of knee surgery, secondary osteoarthritis of the knee, a significant history of knee injury, active knee inflammation, recent intra-articular knee injections within the past 6 months, known hypersensitivity to any study components, inadequate blood examination results or liver function, immune-compromised patients (such as those with HIV infection or diabetes), those at risk for cancer or with a cancer diagnosis, and patients with abnormal blood clotting or those taking anticoagulant medications. Additionally, individuals who had participated in any other interventional study within 4 weeks prior to the start of this study were also considered ineligible. Patients were discontinued from this research study if they experience severe inflammation that could not be alleviated by cold compresses and anti-inflammatory medication. However, it was recorded in this study if additional anti-inflammatory medication was administered.

Participants were required to abstain from any treatments directed at the knee area and from using analgesics, including nonsteroidal anti-inflammatory drugs (NSAIDs), for at least 2 weeks before this study commenced. One participant was excluded during screening due to having a risk of cervical cancer. This study was planned as a prospectively randomized, single-blind (data collector radiologist and interventionist), and active-control study to evaluate the safety and efficacy of autogenic ADSCs. To ensure the high quality of the final cell product, all ADSC preparation processes were conducted at an accredited cell bank laboratory, where a well-established standardized protocol for stem cell isolation and culture was followed. When the cell preparation was completed, the treatment date was scheduled. The study participants were divided into two groups using computer-generated randomization blocks of six. The randomization process was conducted using sealed envelopes, and the numerical codes for each group were generated by a computer. A total of 23 participants (48.94%) had ADSCs harvested by liposuction, purified, and amplified before reinjecting intraarticularly, and 24 participants (51.06%) were injected with hyaluronic acid (control group). The CONSORT flow chart is shown in Figure 6. Patients with knee osteoarthritis were intra-articularly injected by using the ultrasound-guided technique. This study was registered on 29 September 2023 with the Thai Clinical Trials Registry (https://www.thaiclinicaltrials.org/show/TCTR20230929002 accessed on 1 July 2025). The procedures followed were in accordance with the ethical standards of the responsible committee on human experimentation (The Institutional Review Board, Royal Thai Army Medical Department, Phramongkutklao College of Medicine (IRBRTA637/2564) and human research ethic committee, Suranaree University of Technology (EC-64-90), Thailand).

### 4.2. ADSC Preparation

In the treatment group, autogenic ADSCs were harvested and expanded to a quantity of 50 million cells. Enhanced joint function and decreased pain were noted in patients undergoing a bone marrow concentrate protocol, irrespective of the cellular dose.

These ADSCs were prepared in accordance with the standards set by the International Society for Cell & Genetic Therapy (ISCT) and subsequently reimplanted within a one-month timeframe.

ADSCs were prepared from the subcutaneous fat tissue through liposuction of each patient. Specimens were then transported to the Medeze stem cell laboratory in normal saline solution filled with gentamicin antibiotic. Isolation started with placing specimens into sterile conical tube with collagenase to digest the adipose tissue for 45 min at 37 °C. Separation of cells from adipose tissue was performed through centrifugation. Adipose supernatant was removed and ADSCs were placed into 6-well plates with culture media 2 mL/well. ADSCs were then cultured at 37 °C, 5% CO_2_ in alpha-mem medium (Cytiva HyClone, Logan, UT, USA) supplemented with autologous serum for 9 days. They were then trypsinized (0.5% Trypsin -EDTA, no phenol red, Gibco, Grand Island, NY, USA) and reseeded in 4 T75 flasks. The 4 flasks were further cultured for 7 days and re-trypsinized. This time, we reseeded into 16 T-75 flasks. On the day of transplantation, cells were trypsinized and washed four times with saline, then passed through filters. Cells were sampled and counted with an automatic cell counter machine (Countess 3 Cell Counters, Invitrogen, Walham, MA, USA). ADSCs were prepared and resuspended in 5 mL saline in glass vials. The stem cell characteristics of the collected cells were confirmed with flow cytometry using CD73, CD90, CD105 for positive antibodies, and CD45, HLA-DR for negative antibodies against stem cells, respectively. Endotoxin, bacterial contamination, and mycoplasma were tested negative before transplantation.

### 4.3. ADSCs and HA Therapy

The injection materials, whether ADSCs or HA, were prepared by another physician. The syringes were covered with opaque material, ensuring that the injecting physician could not see the preparation process (Appendix A). One syringe (3 mL) of cross-linked hyaluronic acid (LG Chem, Ltd., Seoul, South Korea) was used per patient. The injection was administered using ultrasound-guided techniques [30]. The interventionist who performed the injections was not aware of which treatment the patient received, either ADSC or HA. Following the injection, the evaluators of the WOMAC score, ultrasound, and MRI assessments were all blinded.

The assessor who evaluated the patient’s response to treatment such as ultrasound [31], MRI radiologist, was also blinded. Therefore, the assessor did not know which treatment the patient received, either ADSC or HA. This was carried out to ensure that the assessor’s evaluation was not biased.

The primary outcome was cartilage regeneration, assessed using MRI and ultrasound to measure cartilage and synovial thickness in all participants. Secondary outcomes included the Western Ontario and McMaster Universities Osteoarthritis Index (WOMAC) scores, clinical evaluation, and monitoring of urine CTX-II as a biological marker. Participants were monitored for any adverse events within 6 months of treatment.

### 4.4. MRI Protocol

MRI of the knee was performed using MRI (Philips Achieva 3.0T X-series, Philips Healthcare, Best, NL, USA), utilizing a multi-sequence protocol designed for high-resolution joint evaluation. The imaging protocol included T2-weighted sequences to visualize fluid-sensitive structures and cartilage defects, T1-weighted sequences for bone and anatomical detail, proton density (PD)-weighted sequences to enhance soft tissue contrast, and fat-suppressed sequences using SPAIR (spectral attenuated inversion recovery) for detecting bone marrow edema and synovial inflammation. This combination allowed for detailed assessment of articular cartilage and surrounding structures. To quantify cartilage lesion areas, MRI data were post-processed using 3D Slicer, an open-source image analysis platform. Manual and semi-automatic segmentation tools within 3D Slicer (http://www.slicer.org accessed on 15 July 2025) were employed to delineate cartilage boundaries and calculate lesion areas, supporting objective evaluation of treatment response.

### 4.5. Questionnaires

Participants answered the Western Ontario and McMaster Universities Osteoarthritis Index (WOMAC) pain (range: 0–20) and physical function (range: 0–68) subscales concerning their knee symptoms, respectively, during the last 48 h.

### 4.6. Urine CTXII Assay

The urine samples were collected from patients in both groups before and after treatment at 6 months and stored at 4 °C until analysis. Urinary CTX-II concentrations were measured by using a competitive ELISA test (Urine CartiLaps^®^ (CTX-II) EIA; Immunodiagnostic Systems Ltd. (IDS), Boldon, UK) according to the manufacturer’s instructions. CTX-II levels were normalized to urinary creatinine concentration using the following formula: corrected CTX-II value (ng/mmol Cr)  =  1000 × urine CartiLaps (μg/L)/creatinine (mmol/L).

### 4.7. Statistical Method

Categorical data were described by frequency and percentage. Mean and standard deviation (SD) was applied for normally distributed numerical data, median and interquartile range (IQR) for non-normally distributed numerical data. The normality of data was justified based on histogram and Shapiro–Wilk test. The comparison between dependent normally distributed numerical data was compared by paired T-test. The comparison between independent non-normally distributed numerical data was compared by Mann–Whitney U-test. The comparison between 2 groups of repeated measured variables was applied by generalized linear model for repeated measurement. *p*-values less than 0.05 were considered statistically significant. The analysis was performed using the intention-to-treat principle [32]. All statistical analyses were conducted using Stata version 18 [33].

## 5. Conclusions

This study provides compelling evidence that ADSC therapy is an effective and safe treatment for early-stage knee osteoarthritis. By employing a minimally invasive technique with ultrasound-guided injections, we achieved significant improvements in pain, function, and structural cartilage integrity. Our findings suggest that precise administration of ADSCs near affected cartilage regions enhances treatment efficacy. Additionally, this study underscores the therapeutic benefits of stem cell injections, particularly in patients with early-stage cartilage degeneration. The administration of 50 million ADSCs was effective in reducing cartilage defect size without any reported adverse events. Notably, the observed efficacy appears to be more closely associated with appropriate patient selection, specifically individuals with early osteoarthritic changes, highlighting the importance of initiating regenerative therapy during the early stages of disease progression. Overall, these results support the potential of ADSC therapy as a promising alternative to traditional OA treatments. Further large-scale long-term studies are warranted to optimize treatment protocols, confirm durability of effects, and assess cost-effectiveness in clinical practice.

## Figures and Tables

**Figure 1 ijms-26-08476-f001:**
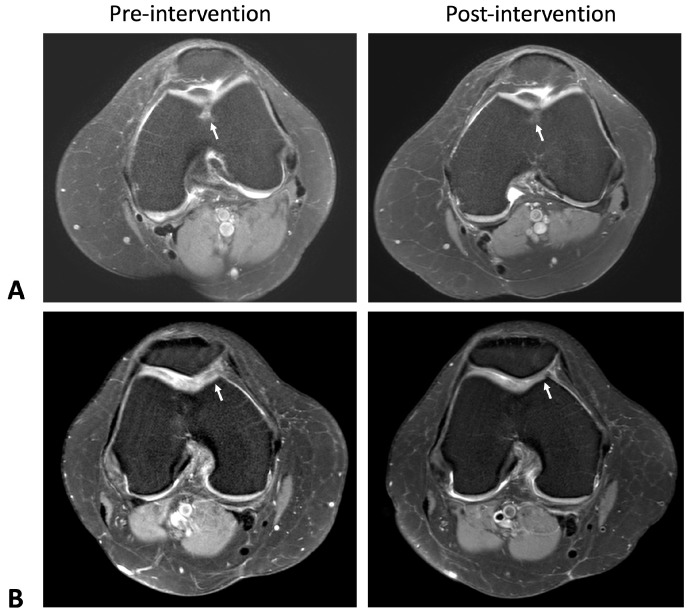
Axial knee MRI images obtained using the T2-weighted echo spin technique, illustrating pre-intervention and post-intervention subjects. (**A**) The arrow indicates a cartilage lesion at the femoral trochlea groove, modified from Outerbridge Grade 3 to Grade 2, with a decrease in subjacent marrow edema. (**B**) Grade 3 cartilage lesion at the medial patella facet ridge reduced in size from 12 × 10 mm to 10 × 9 mm, with a decrease in subjacent marrow edema as indicated by the arrow.

**Figure 2 ijms-26-08476-f002:**
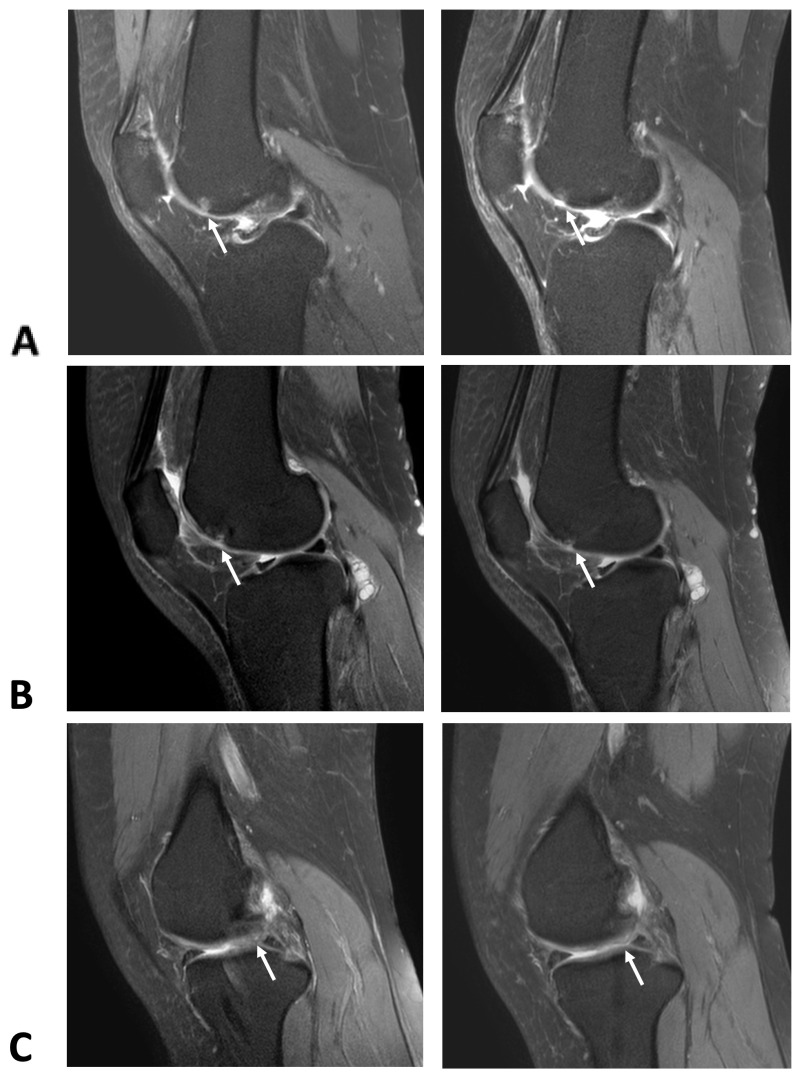
MRI T2-weighted spin echo sagittal view of pre-intervention and post-intervention subjects. (**A**) Arrow indicates a reduction in size of the Grade 2 cartilage lesion at the medial femoral cartilage, from 16 × 12 mm to 15 × 11 mm, along with a decrease in the subjacent marrow edema. (**B**) Arrow shows the disappearance of the previous Grade 1 cartilage lesion at the mid-lateral femoral condyle and a reduction in the subjacent bone edema. (**C**) Arrow indicates a slight decrease in the area of Grade 3 cartilage damage at the medial femoral condyle and medial tibial plateau, measuring 14 × 9 mm and 10 × 6 mm, respectively.

**Figure 3 ijms-26-08476-f003:**
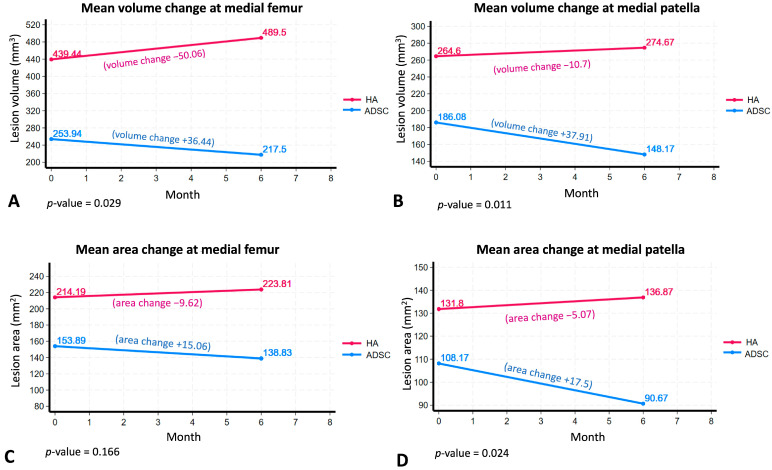
MRI Findings of a mean medial compartment cartilage. Medial cartilage lesion volume in both femur (**A**) and patella (**B**) significantly decreased in the ADSC group but increased in the HA group. Medial cartilage lesion area in both femur (**C**) and patella (**D**) decreased in the ADSC group, whereas the HA group observed more cartilage loss. The blue and pink lines indicate the ADSC and HA groups, respectively.

**Figure 4 ijms-26-08476-f004:**
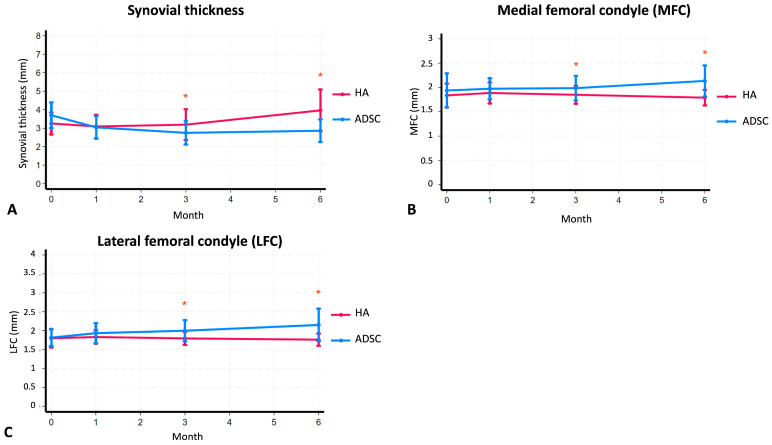
Comparison of (**A**) the thickness of knee synovial tissue, (**B**) the thickness of the medial femoral condyle (MFC), and (**C**) the thickness of the lateral femoral condyle (LFC) between the groups that received hyaluronic acid (HA) and adipose-derived mesenchymal stem cells (ADSCs) after a 6-month period of injection. Asterisks show significant differences between groups at *p* < 0.05.

**Figure 5 ijms-26-08476-f005:**
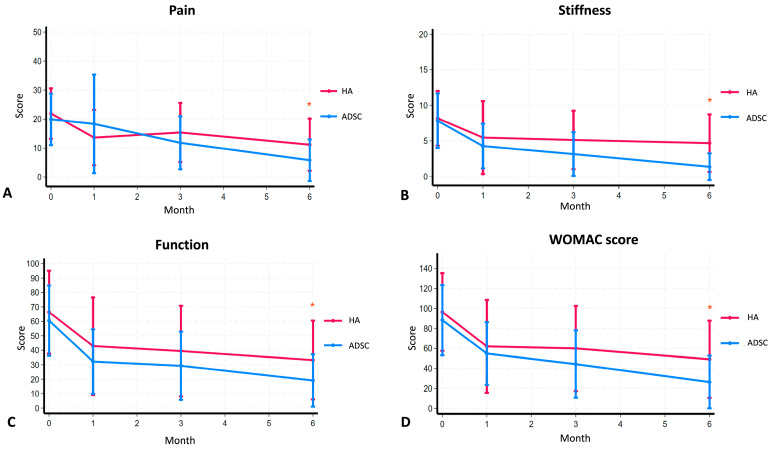
Comparison of (**A**) the pain scores, (**B**) the stiffness scores, (**C**) the functional ability scores from the WOMAC score, and (**D**) the total WOMAC score between the groups that received hyaluronic acid (HA) and adipose-derived mesenchymal stem cells (ADSCs) after a 6-month period of injection. The mean ± SD values are plotted. Asterisks show significant difference between groups at *p* < 0.05.

**Figure 6 ijms-26-08476-f006:**
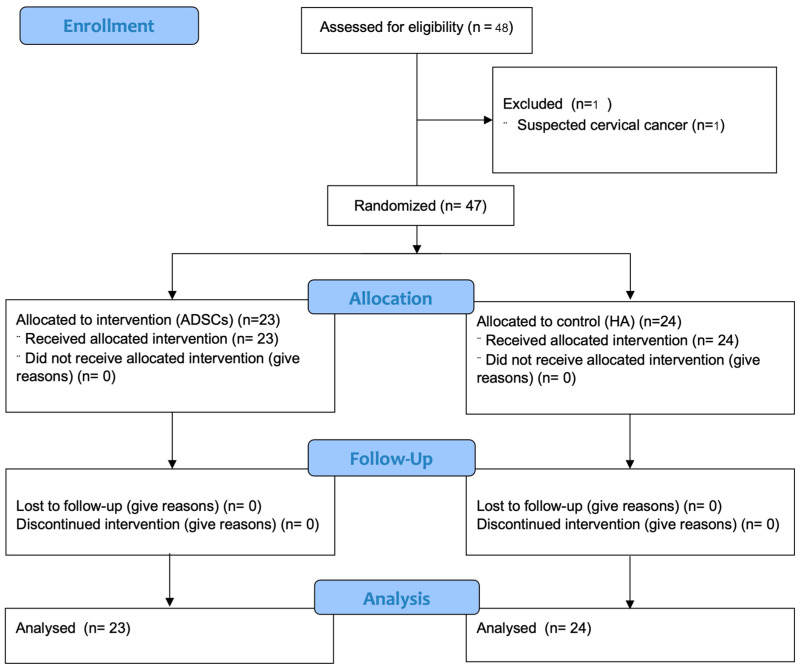
CONSORT diagram. The flow of participants through this study.

**Table 1 ijms-26-08476-t001:** Demographic and baseline data.

Characteristics	HA (Control) (n = 24, 51.06%)	ADSC (Intervention) (n = 23, 48.94%)	*p*-Value
**Gender** (n, %)			
Male	4, 8.51%	4, 8.51%	
Female	20, 44.55%	19, 40.43%	
**Age**	59.00 ± 6.69	56.91 ± 6.15	0.14
**Weight (kg)**	61.23 ± 8.97	61.73 ± 8.62	0.61
**Height (cm)**	158.08 ± 7.85	156.65 ± 5.43	0.24
**BMI (kg/m^2^)**	24.31 ± 2.33	25.12 ± 2.99	0.84
**WOMAC score**			
Pain	21.92 ± 8.70	19.91± 8.84	0.22
Stiffness	8.17 ± 3.84	7.83 ± 3.83	0.38
Function	66.38 ± 28.68	60.52 ± 24.30	0.23
Overall WOMAC score	96.46 ± 38.98	88.26 ± 35.21	0.23
**US**			
Synovial thickness (mm)	3.25 ± 0.60	3.70 ± 0.70	0.98
MFC (mm)	1.84 ± 0.24	1.94 ± 0.35	0.87
LFC (mm)	1.80 ± 0.24	1.81 ± 0.22	0.60
**Urine CTX (ng/mmol)**	310.45 (145.37–566.75)	263 (121.53–486.43)	0.53

MFC—medial femoral condyle, LFC—lateral femoral condyle.

**Table 2 ijms-26-08476-t002:** Compare the thickness of knee synovial tissue before and after injecting adipose-derived mesenchymal stem cells (ADSCs) (n = 23) and after injecting hyaluronic acid (HA) (n = 24) at 1 month, 3 months, and 6 months.

Study Outcomes	Before ADSCs	After Injected ADSCs						Before HA	After Injected HA					
	0 m	1 m	*p*-Value ^a^	3 m	*p*-Value ^b^	6 m	*p*-Value ^c^	0 m	1 m	*p*-Value ^a^	3 m	*p*-Value ^b^	6 m	*p*-Value ^c^
Synovial thickness	3.70 (95%CI 3.40–4.00)	3.04 (95%CI 2.78–3.31)	<0.001 *	2.75 (95%CI 2.48–3.02)	<0.001 *	2.86 (95%CI 2.60–3.12)	0.001 *	3.25 (95%CI 3.00–3.51)	3.09 (95%CI 2.82–3.36)	0.320	3.19 (95%CI 2.84–3.54)	0.756	3.96 (95%CI 3.46–4.46)	0.016 *
MFC (mm)	1.94 (95%CI 1.79–2.09)	1.97 (95%CI 1.88–2.07)	0.612	1.99 (95%CI 1.88–2.10)	0.634	2.13 (95%CI 2.00–2.27)	0.099	1.84 (95%CI 1.73–1.94)	1.89 (95%CI 1.80–1.98)	0.261	1.85 (95%CI 1.77–1.93)	0.765	1.79 (95%CI 1.72–1.86)	0.298
LFC (mm)	1.81 (95%CI 1.72–1.91)	1.93 (95%CI 1.81–2.05)	0.053	2.00 (95%CI 1.88–2.12)	0.020 *	2.15 (95%CI 1.96–2.33)	0.002 *	1.80 (95%CI 1.69–1.90)	1.83 (95%CI 1.75–1.90)	0.539	1.79 (95%CI 1.72–1.86)	0.934	1.76 (95%CI 1.69–1.83)	0.608

^a^ Paired *t*-test 0 m and 1 m, ^b^ paired *t*-test 0 m and 3 m, ^c^ paired *t*-test 0 m and 6 m, * *p* < 0.05. MFC—medial femoral condyle, LFC—lateral femoral condyle.

**Table 3 ijms-26-08476-t003:** Compare the WOMAC score before and after injecting adipose-derived mesenchymal stem cells (ADSCs) (n = 23) and after injecting hyaluronic acid (HA) (n = 24) at 1 month, 3 months, and 6 months.

Study Outcomes	Before ADSCs	After Injected ADSCs						Before HA	After Injected HA					
	0 m	1 m	*p*-Value ^a^	3 m	*p*-Value ^b^	6 m	*p*-Value ^c^	0 m	1 m	*p*-Value ^a^	3 m	*p*-Value ^b^	6 m	*p*-Value ^c^
Pain	19.91 (95%CI 16.09–23.73)	18.35 (95%CI 11.02–25.68)	0.711	11.83 (95%CI 7.89–15.76)	0.007 *	5.83 (95%CI 2.73–8.93)	<0.001 *	21.92 (95%CI 18.24–25.59)	13.63 (95%CI 9.60–17.65)	0.004 *	15.38 (95%CI 11.08–19.67)	0.045 *	11.18 (95%CI 7.21–15.16)	<0.001 *
Stiffness	7.83 (95%CI 6.17–9.48)	4.26 (95%CI 2.90–5.62)	0.003 *	3.13 (95%CI 1.80–4.46)	0.001 *	1.35 (95%CI 0.54–2.16)	<0.001 *	8.17 (95%CI 6.54–9.79)	5.46 (95%CI 3.29–7.62)	0.025 *	5.13 (95%CI 3.39–6.86)	0.028 *	4.68 (95%CI 2.89–6.47)	0.002 *
Function	60.52 (95%CI 50.02–71.03)	32.21 (95%CI 22.59–41.84)	0.002 *	29.30 (95%CI 19.11–39.50)	<0.001 *	19.13 (95%CI 11.29–26.97)	<0.001 *	69.14 (95%CI 56.56–81.72)	42.88 (95%CI 28.65–57.10)	0.005 *	39.42 (95%CI 26.19–52.64)	0.006 *	33.23 (95%CI 21.15–45.31)	<0.001 *
WOMAC	88.26 (95%CI 73.03–103.49)	54.83 (95%CI 41.26–68.39)	0.009 *	44.26 (95%CI 29.66–58.86)	<0.001 *	26.30 (95%CI 14.91–37.70)	<0.001 *	96.46 (95%CI 80.00–112.92)	61.96 (95%CI 42.32–81.60)	0.004 *	59.92 (95%CI 41.91–77.92)	0.009 *	49.09 (95%CI 31.95–66.23)	<0.001 *

^a^ Paired *t*-test 0 m and 1 m, ^b^ paired *t*-test 0 m and 3 m, ^c^ paired *t*-test 0 m and 6 m, * *p* < 0.05.

**Table 4 ijms-26-08476-t004:** Compare the change in urine CTX-II levels over a 6-month period between the injection of adipose-derived mesenchymal stem cells (ADSCs) and hyaluronic Acid (HA).

Study Outcomes	ADSCs	HA	*p*-Value ^a^
Mean change urine CTX-II (ng/mmol) within 6 months	−82 (95%CI −219 to +119.01)	−69.63 (95%CI −214.44 to +132.36)	0.8808

^a^ Mann–Whitney U-test.

## Data Availability

All data generated or analyzed in this study are available within this published article. For additional information or requests, please contact the corresponding authors.

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
