# Peer review of "Cartilage Regeneration Potential in Early Osteoarthritis of the Knee: A Prospective, Randomized, Open, and Blinded Endpoint Study Comparing Adipose-Derived Mesenchymal Stem Cell (ADSC) Therapy Versus Hyaluronic Acid"

_ijms, 2025, doi:10.3390/ijms26178476_

Round 1
Reviewer 1 Report
Comments and Suggestions for Authors
- Type in full (adipose-derived mesenchymal stem cells) in the title of your paper so its clear what your study is about.
- Page 2, line 58, should say "adipose-derived"...fix everywhere.
- Page 2, line 64, sentence needs to be reworded as it just ends.
- Page 2, line 71, delete "study" as its repetitive.
- Page 2, line 72, should say "have" not has.
- Page 2, line 72, got to have a capital letter to begin a sentence.
- Page 3, line 92, should say "measured by WOMAC scores"
- Page 3, lines 95-102 of results...not clear on what groups are being reported on in brackets and vs words...make clear.
- Page 3, line 104, Table should say "Demographic and baseline Data" its not all demographic, there are results there as well.
- Page 5, line 38, space between marrow and edema needed.
- Page 6, line 162, spelling is patella not patellar....fix everywhere.
- Page 7, Figure labelling, synovial thikness? fix.
- Page 7, line 190, should say Asterix shows significant differences.
- Tables 2 and 3 seem to have a lot of T-test comparisons, another statistic could have been better like a RM-ANOVA
- Page 13, line 340, should say "formula"
- Page 15, line 410, just checking whether this study was single or double blinded?
- Page 15, line 442, why are patients answering WOMAC on hand and hip issues? clarify.
- overall comment; not sure why you have put results before methods...should be intro, methods, results, discussion, refs...please reorder paper.
- A few of your references have capitals in their title...fix.
- Page 15, line 431, you have written "test" twice.
The quality of the English Language was of a low quality and needs to be improved on a revised version. Lots of spelling and grammar issues.
Reviewer 2 Report
Comments and Suggestions for Authors
The manuscript by Tangkankanavelukul et al. focusses on a clinical study on cartilage regeneration potentail in early osteoarthritis of the knee, by comparing ADSCs cell- versus HA-therapy in two randomised partient groups (48 patients in total). Both therapy approaches were administered intra-articular into the knee by ultrasound guide technique, and the effects were evaluated over a six-month period after intervention.
The data of the study presented suggest that ADSCc injections result in better cartilage restoration, and overall reduced symptoms and improvements in WOMAC scores, even though HA treatment also showed reduced symptoms, but not decreased cartilage lesion volume. However, the reviewer has several issues that need to be addressed before the manuscript can be published:
1) Introduction: Background information and state-of -literature on HA and medical treatment is completely missing. The authors are asked to include the relevant information and refs.
2) Figures: Figs. 3-5 (p.6-8) are too small, and the axis labelling is hardly readible. In Fig. 3 (p.6), the assignment of the groups and colours seems to be flipped in the description l.165. Please correct to be unambiguously.
In Fig. 1B (p.4), the reviewer could not follow the statement of "decrease in edema", since the MRI images look pretty much the same. Please comment on this.
3) Results p.11, l.240: The given results/calculations of urine CTX-II are not presented in a clear and logial way; the calculations of the value decreases do not fit to the values in the brackets. In addition, the referenced Table 4 is not provided for the Biomarker analysis. Please rewrite this paragraph and add the Table to the manuscript, if necessary.
4) Discussion p. 12, l. 307: According to the results presented in Table 3 (p. 10), the function core / WOMAC scores should either decrease, or improve in the cited ref [18], not increase. Please correct accordingly.
5) Materials & Methods: p.13, l.358 Even though the preparation of the ADSC cells was conducted at an accredited cell bank laboratory, this section misses the infos of the protocol for cell isolation and culture. The authors should at least include a short version of the steps, chemicals & buffers. Information for HA gel production and source is missing. Infos for MRI device and image analysis procedure is missing. The authors are asked to add the missing information.
Reviewer 3 Report
Comments and Suggestions for Authors
Tangkanjanavelukul et al. is a well designed study with important merit for treatment of early osteoarthritis. The authors used appropriate blinding techniques to avoid bias and collected meaningful data which suggests that adipose-derived mesenchymal stem cells are a viable treatment for mild osteoarthritis. The data are not over-interpreted and the authors are commended for including a description of the study limitations.
Major points:
- Table 4 is missing.
- All data figures are too small and with too poor contrast for a printed copy. They would only be ok on a screen where they may be zoomed into. Please increase in size, including font size, and provide more contrast.
- In the Methods, please include a description of how urine CTX-II was measured.
- In the Conclusions on lines 442 to 445, you write "... this study highlights the dose-dependent effects of stem cell therapy, demonstrating that a mid-dose of 50 million cells was sufficient to reduce cartilage defects without adverse events. This finding contrasts with previous studies using various dosages, which often led to inconsistent outcomes." I am confused about this claim, as your study did not include any other doses but the 50 million cells, which you note as being a mid-dose. This suggests to me that a dose response curve may have been completed. If so, the data should be presented as a supplementary file if not in the results. May I recommend that the short discussion on the dose-dependency should be moved into the body of the discussion with appropriate references and data to support this claim. Or, perhaps these sentences should be clarified in the Conclusion if I have misunderstood the point you are making.
Minor points:
- "ADSCs cell therapy" (Title, line 4) is redundant; should the abbreviation be written out fully? If not, then suggest writing "ADSC therapy".
- Should WOMAC be written out at first use in the abstract (line 27)?
- Use of numerals vs written numbers is inconsistent. Most numbers are written out (e.g., 'six', for example), but in the abstract and methods, numerals are used instead (e.g., 6). Please be consistent.
- Use of abbreviations is not consistent. Please introduce abbreviation at first use and then use the abbreviation within the text thereafter.
- "Despite the use of stem cells, with inclusion of both early-stage and advanced-stage knee osteoarthritis patients." (lines 64 and 65) is an incomplete sentence.
- Table 1 and 2: suggest either write out "MFC" and "LFC" in the table or provide an explanation for the abbreviations below the table.
- Table 3 is missing the explanation of superscripts a - c.
- Figures 1 and 2: would representative images from the hyaluronic acid group be meaningful comparisons?
- Line 122 reads "...28.44 mm squared (95% CI..." - what is the CI? No values are provided for this range... unless I misunderstood this.
- Line 156: "(p<" not "(p=<".
- Lines 217 to 223 do not refer to any figures or tables.
- Figure 5: it would help if the title included the information that mean +/- SD are plotted.
- Discussion: while the data are reiterated, the relevant figures and/or tables are not referred to.
- Line 308: please write out what "KOOS" stands for.
- Lines 351 and 352: the sentence should be past tense.
- Figure 6: should "(give reasons)" be removed?
- Line 384: is "(50 M)" necessary? It is confusing as "M" is also the abbreviation for molar concentration.
- While the supplementary material includes an equation for calculating sample size with sufficient statistical power, this is not mentioned in the section on statistical method (lines 424 to 434. Please do so or at least refer to your supplementary material.
- References in the methods section would be helpful. For example, on line 434, "intention-to-treat principle".
- The list of abbreviations is not complete. For example, KOA is not included. However, it might be better to use "knee OA" through-out the paper rather than introduce "KOA" late in the paper, for consistency.
The manuscript reads well and is carefully prepared but still contains numerous grammatical and other mistakes. It needs to be carefully proof-read and corrected.
Round 2
Reviewer 2 Report
Comments and Suggestions for Authors
Dear authors, thank you for the manuscript revision, which quality has been significantly improved. All comments by the reviewer have been satisafctory addressed, and the interesting paper may now be ready for publication.